# Unipolar quantum optoelectronics for high speed direct modulation and transmission in 8–14 μm atmospheric window

Hamza Dely [1,8] ✉, Mahdieh Joharifar[2,8], Laureline Durupt[3,8], Armands Ostrovskis[4], Richard Schatz[2], Thomas Bonazzi[1], Gregory Maisons[3], Djamal Gacemi[1], Toms Salgals[4], Lu Zhang[5], Sandis Spolitis[4], Yan-Ting Sun [2], Vjačeslavs Bobrovs [4], Xianbin Yu[5], Isabelle Sagnes [6], Konstantinos Pantzas [6], Angela Vasanelli[1], Oskars Ozolins[2,4,7], Xiaodan Pang [2,4,7] ✉ & Carlo Sirtori [1] ✉

The large mid-infrared (MIR) spectral region, ranging from 2.5 μm to 25 μm, has remained under-exploited in the electromagnetic spectrum, primarily due to the absence of viable transceiver technologies. Notably, the 8–14 μm long-wave infrared (LWIR) atmospheric transmission window is particularly suitable for free-space optical (FSO) communication, owing to its combination of low atmospheric propagation loss and relatively high resilience to turbulence and other atmospheric disturbances. Here, we demonstrate a direct modulation and direct detection LWIR FSO communication system at 9.1 μm wavelength based on unipolar quantum optoelectronic devices with a unprecedented net bitrate exceeding 55 Gbit s⁻¹. A directly modulated distributed feedback quantum cascade laser (DFB-QCL) with high modulation efficiency and improved RF-design was used as a transmitter while two high speed detectors utilizing meta-materials to enhance their responsivity are employed as receivers; a quantum cascade detector (QCD) and a quantum-well infrared photodetector (QWIP). We investigate system tradeoffs and constraints, and indicate pathways forward for this technology beyond 100 Gbit s⁻¹ communication.

Driven by growing bandwidth demands, wireless communications are transitioning from microwaves to millimeter-waves (MMW) and soon to terahertz (THz). This trend points towards an all-spectra communication paradigm, utilizing any available electromagnetic (EM) resources across radio and optics for bandwidth[1]. Mid-infrared (MIR) (3–30 μm) represents a compelling segment of the EM spectrum, raising significant interest for applications in spectroscopy[2–10], defense[11–13], astronomy[14,15],

and free-space optical (FSO) communications[16,17]. Two atmospheric transmission windows in the MIR, namely, the mid-wave infrared (MWIR, 3–5 μm) and the long-wave infrared (LWIR, 8–14 μm), hold intrinsic advantages for both terrestrial and space applications. They provide broader bandwidth and nearly 100 times lower atmospheric water absorption than MMW and THz[17]. They also experience considerably reduced Mie scattering, commonly found in meteorological

[1]Laboratoire de Physique de l'ENS, Département de Physique, École Normale Supérieure, Université PSL, Sorbonne Université, Université Paris Cité, CNRS, 75005 Paris, France. [2]Department of Applied Physics, KTH Royal Institute of Technology, 106 91 Stockholm, Sweden. [3]mirSense, 2 Bd Thomas Gobert, 91120 Palaiseau, France. [4]Institute of Telecommunications, Riga Technical University, 1048 Riga, Latvia. [5]College of Information Science and Electrical Engineering, Zhejiang University, Hangzhou 310027, China. [6]Université Paris-Saclay, CNRS, Centre de Nanosciences et de Nanotechnologies, 91120 Palaiseau, France. [7]RISE Research Institutes of Sweden, 164 40, Kista, Sweden. [8]These authors contributed equally: Hamza Dely, Mahdieh Joharifar, Laureline Durupt. ✉ e-mail: hamza.dely@ens.fr; xiaodan@kth.se; carlo.sirtori@ens.fr

phenomena such as dust, haze, and low-altitude clouds, than the 1.55-μm telecom band[18]. Therefore, MIR potentially offers high link availability through the atmosphere[19].

Two main MIR FSO communication approaches are wavelength conversion and direct-emitting sources[20]. The former, by employing nonlinear parametric conversions, leverages mature fiber-optic components, supports very high data rates, offers compatibility with fibre-optic systems, and facilitates multi-dimensional multiplexing[21–24]. The latter focuses more on footprint and energy consumption to reinvent compact MIR FSO transceivers. An early study utilized a direct-emitting PbCdS diode laser at 3.5 μm for 100 Mb/s data transmission, with speed limited by carrier lifetime[25]. Since the 1990s, there has been an advent of a new assemblage of optoelectronic devices based on intersubband transitions, such as quantum cascade laser (QCL)[26], quantum-well infrared detector (QWIP)[27] and quantum cascade detector (QCD)[28–30]. Using III-V semiconductor heterostructures, these unipolar devices can target any wavelength from MIR to THz[31]. They bring opportunities to build novel integrated systems for sensing and communication[32,33]. In particular, data transmission benefits from the intrinsic high-speed properties of intersubband devices due to their fast electron relaxation time under a picosecond[34]. Previous characterizations at the component-level have demonstrated promising results regarding QCL modulation response[35–39], detectors' bandwidth, as well as receiver responsivity[40–44]. These results encourage further explorations at the system level, which is far more complicated to characterize than the single components and devices. The implementation of an operational system present unique challenges arising from the holistic optimizations and evaluations of the overall trade-offs of each component, in terms of bandwidth, power, and linearity under real operating conditions. Furthermore, the characteristics of the optoelectronic components must be synthesized with information and coding theory, along with advanced communication technologies such as modulation and signal processing techniques, to maximize transmission performance in such systems[20]. Even though preliminary data transmission efforts were initiated in the early 2000s, most devices at that time needed to be operated at cryogenic temperature[45,46]. The subsequent two decades have witnessed remarkable advancements in both unipolar lasers and detectors, paving the way for room-temperature operation. To date, several MIR FSO transmissions are carried out using directly modulated QCL[47–51], reaching speed over 10 Gbit s$^{-1}$. However, these devices' potential remains underutilized, mainly due to QCL's suboptimal RF mounting and detectors' low signal-to-noise ratio (SNR) at room temperature. Here, we've optimized the bandwidth of a QCL chip through refined RF design and bonding, and enhanced room-temperature QCD and QWIP's responsivity and speed by combining metamaterials with III-V heterostructures[43]. Utilizing these refined devices, we achieved unprecedented >55 Gbit s$^{-1}$ net bitrate LWIR FSO transmission at room temperature.

## Results

### Characteristics of unipolar quantum optoelectronic devices

Two distributed feedback (DFB) QCLs were designed, fabricated and compared for FSO transmission (see details in Methods). Both lasers originating from the same process flow, emit at around 9.1 μm wavelength and operate in continuous-wave (CW) mode at room temperature. However, the two QCL chips differ in width, wiring, and submount soldering configurations, addressing the tradeoff between power and bandwidth. The first QCL chip, referred to as Standard-QCL, is optimized for high output power through a dedicated thermal dissipation design. It is 2-μm wide, soldered epi-down, and bonded onto a standard submount tailored for DC operation. It is then placed on a copper block with a cooling system, achieving an output power of more than 30 mW in CW mode at 15 °C. In contrast, the second QCL chip, referred to as RF-QCL here, is optimized for RF characteristics. It has a width of 4 μm, and is soldered epi-up onto a cleaved submount of

the exact dimensions as the QCL chip. This cleavage enables laser bonding using short wires for high-speed operation. Then, the QCL chip is soldered onto a copper base plate, followed by bonding onto a printed circuit board (PCB). Additional wirebonds connect the bottom of the laser to the PCB on each side of the central line, as shown in Fig. 1a. The PCB was designed to accommodate an SMA connector for RF injection, utilizing the high-speed capabilities of the QCL. The RF-QCL emits a relatively lower power of 13 mW at 15 °C, mainly due to the thermal dissipation challenge from the suboptimal indium soldering at the interface between the submount and the copper base plate. In summary, the Standard-QCL prioritized output power with optimized heat dissipation, whereas the RF-QCL emphasized bandwidth with refined RF design.

The light-current-voltage (L-I-V) curves for both QCLs are shown in Fig. 1b, measured at 15 °C with mounted FSO setup. The Standard-QCL has a threshold current of around 400 mA, while the RF-QCL's is about 350 mA. The RF-QCL also experiences slight roll-off beyond 460 mA. The normalized spectra of both lasers at different bias points are depicted in Fig. 1c, d, measured at 15 °C with a spectrometer (Nicolet iS50 FTIR). For both lasers, the emission peak shifts to longer wavelengths with higher bias current, as theory predicts. The current-tunning coefficient is calculated to be $(264 \pm 4)$ MHz/mA for the Standard-QCL, and $(193 \pm 10)$ MHz/mA for the RF-QCL. To verify the enhanced modulation bandwidth of the RF-QCL, additional characterizations were performed using electrical rectification[52] (see Methods and Supplementary Information Note 2). Modulation bandwidth of around 10 GHz can be obtained at different bias points with a flat frequency response, with neither relaxation oscillations nor electrical free spectral range (FSR) resonances, as shown in Fig. 1e. This result represents a substantial improvement compared to a previous design[51].

Figure 2a, b present SEM images of the QCD, which layout is analogous to the QWIP. The device fits in a $60 \times 60\ \mu m^2$ area. A 50 Ω coplanar waveguide and an air bridge are processed on the sample for easier electrical connection and enhanced frequency performance. These home-made detectors use a one-dimensional stripe array metamaterial design where the active region (AR) is embedded in stripe-shaped metal-semiconductor-metal resonators, with width $S$ matching half the AR's resonant radiation wavelength (Fig. 2c). Conceptually, metamaterial-based intersubband detectors function like two coupled oscillators, the cavity and the intersubband transition, exchanging energy[53,54]. S is chosen so that the cavity resonant frequency matches the AR peak absorption wavelength for each device. The resonator height H can be adjusted to modify both the cavity mode's coupling efficiency with incident radiation and its quality factor. Both QCD and QWIP operate in the weak light-matter coupling regime.

Stripe-array detectors, whose layout is depicted in Fig. 2d, have three advantages. First, they confine the incoming EM energy $S_{in}$ in a $TM_{01}$ mode within the subwavelength cavity, intensifying the electric field in the AR. Second, the direction of the electric field $\vec{E}_{cav}$ becomes vertical, satisfying polarization selection rules for intersubband transitions. Lastly, they reduce the device's electrical surface, decreasing electric noise and capacitance. The outgoing EM energy $S_{out}$ and the detector absorbed energy can be calculated with the coupled-mode theory[54]. Figure 2e shows responsivity spectra at room temperature of the QCD and the QWIP, with notably more localized peak responses than conventional designs. Specifically, the absorption peak of the QCD (no bias) is found to be around 10 μm (124 meV) with a peak responsivity of 26 mA/W, and for the QWIP (at 1.1 V bias) it is at 9.3 μm (134 meV) with 320 mA/W peak responsivity. The full-width half-maximum of the QCD spectrum is slightly narrower than the QWIP, both around 20 meV. For the transmission experiment, the peak of the QWIP response almost aligns with the emission energy of the two QCLs (see Fig. 1c, d), whereas the QCD is slightly off-resonance. The bandwidth of both detectors are characterized with electrical rectification, and the measurement results are shown in

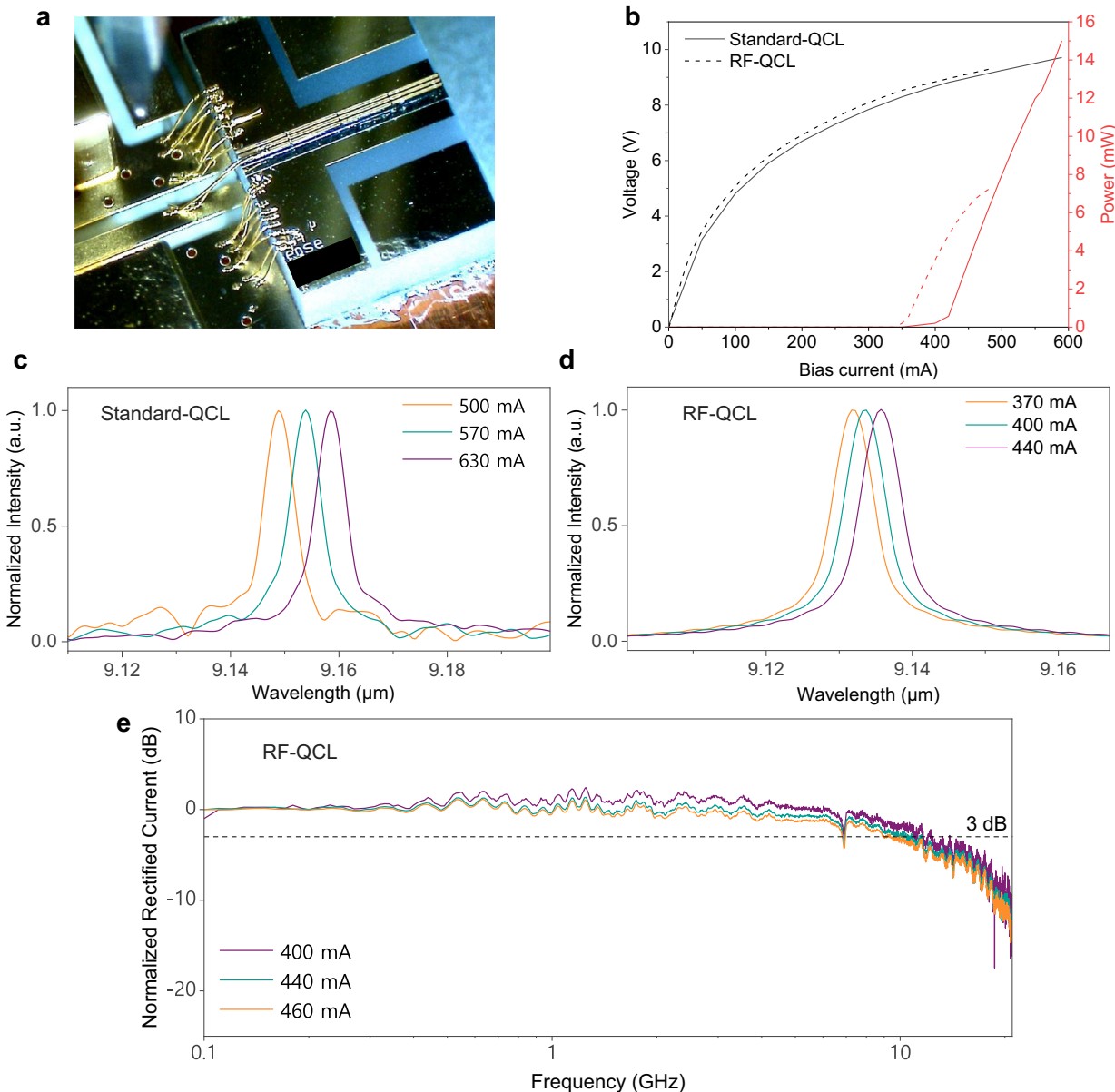

**Fig. 1 | Two QCLs used for FSO transmission and their characteristics. a** A microscopic photo showing the the wirebonding between the cleaved RF-QCL submount and the PCB to enhance the frequency response. **b** The measured L-I-V curves of both the Standard-QCL and the RF-QCL at 15 °C in CW mode. Note that these measurements are performed in the system-level setup, and characterized power values are lower than chip-level measurements due to the beam divergence.

While the actual output power of the QCL chips are higher, these measured values correspond to the actual power levels the detectors receive in the free-space transmission setup. The measured optical spectra with normalized intensity with respect to the bias current of **c** the Standard-QCL, and **d** the RF-QCL. **e** The characterized modulation bandwidth of the RF-QCL with electrical rectification at different bias current points.

Fig. 2f. The QCD has a 3-dB bandwidth of about 12 GHz and a smoother frequency rolloff afterwards, while the QWIP has around 9 GHz bandwidth, beyond which a sharper rolloff is observed. To summarize the detectors' tradeoffs: the QCD has higher bandwidth but lower responsivity, whereas the QWIP provides slightly lower bandwidth but greater responsivity. In addition, the QWIP signal is noisier due to thermally excited electrons being accelerated by the applied voltage, while the unbiased QCD has better noise performances. These tradeoffs in both lasers and detectors lead to the necessity of a thorough transmission performance evaluation.

### FSO transmission setup

The FSO transmission setup is shown in Fig. 3a. Different signal formats, i.e., non-return to zero (NRZ) and multilevel pulse amplitude

modulation (PAM), were generated with an arbitrary waveform generator (AWG). These signals were amplified and then added to the DC bias current through a bias-tee to drive the QCLs. The output LWIR beam was collimated, transmitted through an FSO link, and focused onto the detector. The detector output was amplified and sampled by a real-time digital storage oscilloscope (DSO). Transmission performance was evaluated by bit error rate (BER) after offline digital signal processing (DSP). We determined performance based on the highest achievable symbol rates across various modulation formats and laser-detector combinations. Setup details are described in Methods, and the DSP specifics are described in the Supplementary Information. The end-to-end system response calibrations for all test cases are shown in Fig. 3b, with calibration process details described in Methods and Supplementary Information Note 5.

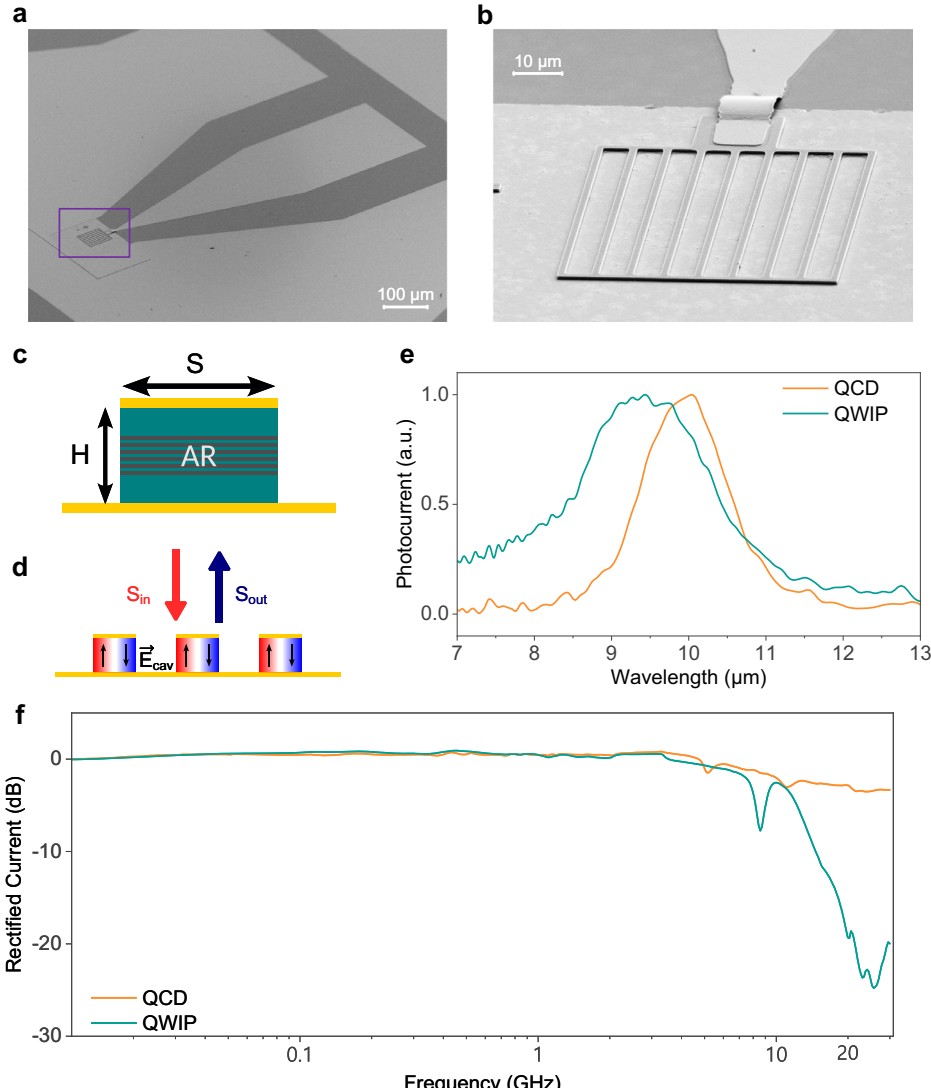

**Fig. 2 | The two photodetectors used for free-space transmission and their characteristics. a** SEM picture of the stripe-based QCD and the view with its coplanar waveguide. The QWIP used in this experiment has the same structure and appearance. **b** The zoomed-in image on the device. Each stripe is 1.5 μm wide and the space between each stripe is 7 μm. Both the QCD and the QWIP are metamaterial-patterned. **c** Lateral view of a single detector stripe cavity resonator with its active region (AR). **d** Sketch of the stripe-array detector with illustrations of the electric field $\vec{\mathbf{E}}_{cav}$ induced by the incident electromagnetic signal $S_{in}$ polarized along the resonant dimension of the stripe. $\vec{\mathbf{E}}_{cav}$ turns vertical to both meet the polarization selection requirement. **e** The normalized photocurrent response spectra for both the QCD (orange), and the QWIP (green). **f** The characterized detection bandwidth of the QCD (orange) and the QWIP (green) with electrical rectification.

## FSO transmission performance with the Standard-QCL

First, we evaluated the Standard-QCL transmission performance and compared the QCD and QWIP. The BER results using the QCD are shown in Fig. 4a. The Standard-QCL's high output power produced a satisfactory SNR, given the QCD's limited responsivity. The system supports up to 33 Gbaud NRZ, 18 Gbaud PAM4, and 13 Gbaud PAM6, meeting the 6.25% overhead hard-decision forward error correction code (HD-FEC) threshold[55] of $4.5 \times 10^{-3}$. This translates to net bitrates of 31.05 Gbit s$^{-1}$ for NRZ, 33.8 Gbit s$^{-1}$ for PAM4 and 30.5 Gbit s$^{-1}$ for PAM6. As shown in Fig. 4b, clear eye diagrams and distinct separations in the symbol distribution are observed. When weighing the system tradeoffs between bandwidth, SNR, and linearity, PAM4 stands out as the most optimal of the three tested modulation formats, as it supports the highest achievable bitrate within this experimental configuration.

We subsequently assessed the performance of the QWIP. Figure 4c, d show the BER results, and received signal eye diagrams with distribution histograms. Notably, the QWIP enabled higher symbol rates across all three modulation formats due to its higher

responsivity. Specifically, transmissions reached 38 Gbaud for NRZ, 21 Gbaud for PAM4, and 15 Gbaud for PAM6, all with BER performances below the HD-FEC threshold. This translated to net bitrates of 35.7 Gbit s$^{-1}$ (NRZ), 39.5 Gbit s$^{-1}$ (PAM4), and 35.2 Gbit s$^{-1}$ (PAM6).

## FSO transmission performance with the RF-QCL

We then switched to the RF-QCL and performed the same system evaluation. We firstly placed the QCD at the receiver and the results are shown in Fig. 5, which reflect the tradeoff between the RF-QCL's enhanced bandwidth and its lower output power. Consequently, as one can observe from Fig. 5a, the system achieves up to 42 Gbaud NRZ with BER below the HD-FEC threshold. This translates to a net bitrate of 39.5 Gbit s$^{-1}$, a result of the enhanced RF performance of the laser. The eye diagram and the symbol distribution histogram are shown in Fig. 5b. In contrast, when we apply PAM4 signaling, due to the limited SNR of the system, we only achieved 5 Gbaud below the HD-FEC limit, i.e., 9.4 Gbit s$^{-1}$ net rate. We'd like to note the potential for a higher symbol rate with this configuration. We pinpointed this rate by

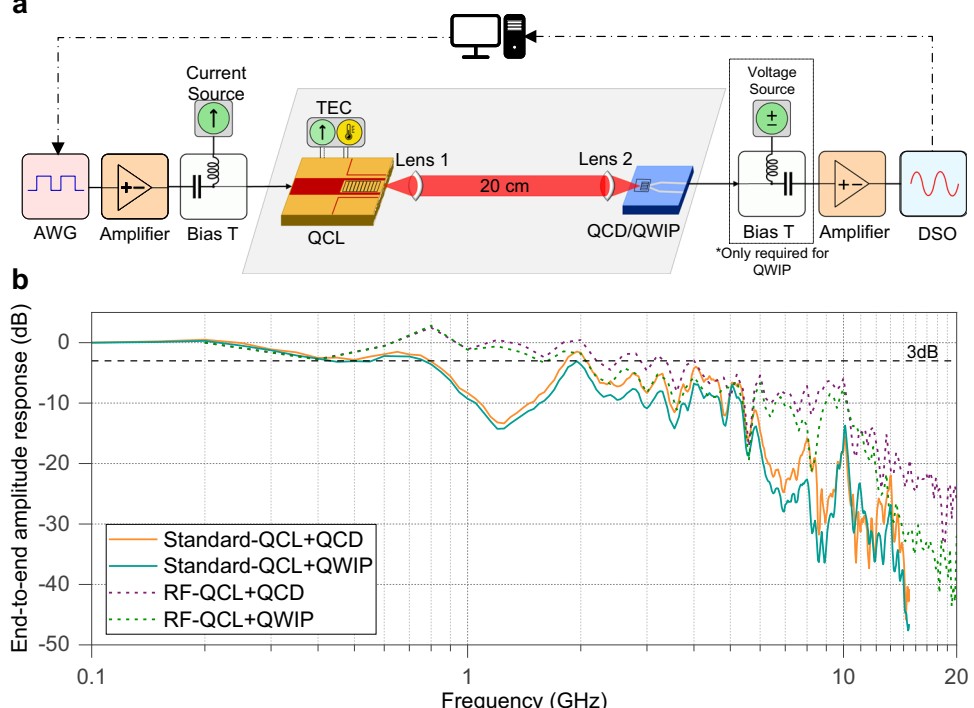

**Fig. 3 | Experimental setup and measured system response. a** The modulated signal is generated offline at a lab computer with MATLAB and loaded to the arbitrary waveform generator (AWG). The AWG output is firstly amplified and then combine with a DC bias current at a high-current bias-tee (2 A, 40 GHz), before driving the QCLs. The QCLs are mounted on a Peltier element with thermoelectric cooling (TEC) to configure and stabilize the operational temperature. Two types of detectors, i.e., QCD and QWIP, are mounted 20 cm away from the QCL mount. A pair of f/1" ZnSe collimation lenses are placed between the laser mount and the detector mount to collect the emitted energy focusing on the detectors. For the QWIP, a second bias-tee with a voltage source is connected to provide the bias voltage, whereas for the QCD no bias voltage is required. It's worth noting that no cooling or temperature control is needed for both types of detectors. The received signal is amplified and captured by a real-time storage oscilloscope (DSO), and the converted digital samples are sent back to the lab computer for demodulation. **b** The system's characterized end-to-end S21 amplitude response, including the cascaded frequency response of all the electrical and optoelectronic devices in the setup.

sweeping over various symbol rates when biased at the maximum current of 480 mA (refer to Supplementary Fig. 12). However, it was later detected that when the bias current exceeded 460 mA, the performance degraded due to modulation nonlinearity. Examining the eye diagram and symbol distribution in Fig. 5c, there's noticeable compression between the top amplitude levels compared to the lower ones, leading to increased bit errors. Another contributing factor is the peak-to-peak voltage of the modulated signal. The QCD's low responsivity requires a higher modulation signal amplitude to combat SNR limitations. This, however, caused the QCL entering modulation non-linearities, i.e., the L-I roll-off region (see Fig. 1b), more quickly with increased bias current. As the system is strictly SNR limited, we didn't test higher modulation levels than PAM4 with the RF-QCL.

Finally, we placed the QWIP at the receiver, which allowed us to achieve the highest bitrate among all tested laser-detector combinations. The BER results of the NRZ and PAM4 transmissions are shown in Fig. 6a, b. For NRZ, the highest achievable symbol rate fulfilling HD-FEC was 55 Gbaud, yielding a net bitrate of 51.7 Gbit s$^{-1}$. We also benchmarked the performance against two other FEC thresholds. The first one has a lower coding gain yet with lower complexity and latency, namely, the Reed–Solomon (RS) (528,514) code[56], here referred to as the KR-FEC limit, which has an overhead of 2.7% and a pre-FEC BER threshold of $2.2 \times 10^{-5}$. This FEC scheme is ideal for short-reach, latency-sensitive scenarios, e.g., terrestrial FSO supporting radio access networks (RAN). The second one, i.e., 20% overhead soft-decision FEC code[57], referred to as the SD-FEC limit in this paper. It has a higher coding gain and a pre-FEC BER limit of $2 \times 10^{-2}$. Such an FEC typically adds complexity and latency to optical transceivers, thus mostly used in long-distance scenarios like ground-to-satellite FSO to avoid re-

transmission. As shown in Fig. 6a, 40 Gbaud NRZ can be transmitted and received with BER below the KR-FEC limit, resulting in a net bitrate of 39.1 Gbit s$^{-1}$. Subsequently, up to 65 Gbaud NRZ can achieve below the SD-FEC limit, achieving a net bitrate of 54.1 Gbit s$^{-1}$. Selected NRZ eye diagrams and symbol distribution histograms for the highest achievable symbol rate for each FEC threshold are displayed in Fig. 6c.

For PAM4, as illustrated in Fig. 6b, we reached a BER below the HD-FEC threshold at 30 Gbaud, corresponding to a net bitrate of 56.4 Gbit s$^{-1}$. When benchmarked against the higher SD-FEC threshold, up to 35 Gbaud PAM4, i.e., 58.3 Gbit s$^{-1}$ net bitrate, can be achieved. This is the highest demonstrated bitrate across all test cases. Further increasing the symbol rate to 40 Gbaud results in an above SD-FEC performance. Selected eye diagrams and symbol distribution histograms for PAM4 are shown in Fig. 6d. Different from the PAM4 results with the QCD as the receiver, negligible nonlinear compression is observed with this configuration at high bias currents. This change is attributed to the superior responsivity of the QWIP, which permits a reduced peak-to-peak voltage for the modulated PAM4 signal, thus preventing the RF-QCL from operating in a nonlinear regime.

## Discussion

Our high-speed LWIR FSO setup relies on two key enabling factors. First, the RF-QCL with optimized high-frequency characteristics enhances the system bandwidth, thus the transmission rate, despite a cost of approximately 3-dB output power compared to the Standard-QCL. Second, metamaterial-enhanced unipolar detectors, i.e., QCD and QWIP, outperform our prior conventional design[51], offering both enhanced SNR and bandwidth owing to their higher responsivity and reduced electrical surface.

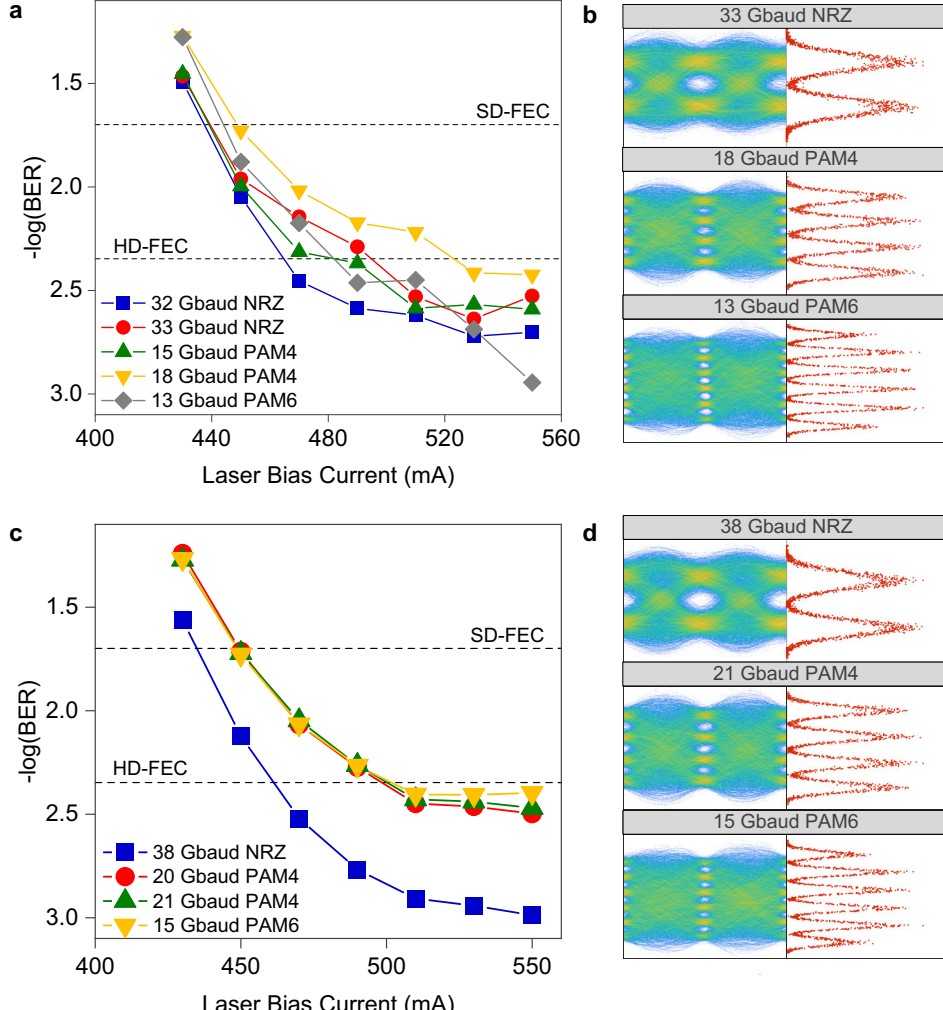

**Fig. 4 | Free-space transmission results using the Standard-QCL. a** BER results with the QCD at the receiver. The BER are measured as a function of the QCL bias current for different modulation formats, namely, NRZ, PAM4, and PAM6. Two FEC code thresholds, i.e., 6.25% overhead HD-FEC of $4.5 \times 10^{-3}$ BER, and 20% overhead SD-FEC of $2 \times 10^{-2}$ BER, are shown to benchmark the performance. **b** Selected eye diagrams of the received signal detected with the QCD, measured at the highest bias current after receiver equalization. The distribution of recovered symbols at the decision points are shown in the histograms. **c** BER results with the QWIP at the receiver. **d** Selected eye diagrams and symbol distribution histograms of the signals detected with the QWIP, measured at the highest bias current after receiver equalization.

There are still potential improvements to be made in the current system. The first is the limited output power of the RF-QCL. The consistent downward trend in the BER curves shown in Fig. 6a, b suggests that the system is still noise limited, suggesting that better results can be expected with higher power. For this experiment, one potential way to increase RF-QCL output power was to lower its operational temperature, which, however, would consume higher TEC power and pose a risk of condensation to damage the QCL chips. We can indeed improve the sealing of the QCL mounting to prevent moisture accumulation, potentially leading to higher output power, and subsequently higher data rates. A more sustainable approach is to enhance the heat dissipation design. The most straightforward approach would be to optimize the indium soldering between the submount and the copper base plate, to allow higher output power without lowering the TEC temperature. However, the performances would be greatly improved with direct soldering of the laser chip on a specifically RF-designed AlN submount which comprises RF connections. This would allow to move away from indium soldering on a copper plate and benefit from industry standard AuSn soldering on the AlN submount.

There are also limitations on the detector side. For the QCD, its high bandwidth merit has been severely hindered by its low responsivity, partly due to the offset between the QCL emitting wavelength and the detector's responsivity peak. And for the QWIP, its relatively lower bandwidth and higher thermal noise level could be further improved to enhance the system performance. One way forward to improve the response of both detectors is to replace the stripes in the current design with a patch array layout, so as to benefit from a reduced electrical area for high speed operation, and also avoid polarization issues[42,43]. Further enhancement of the detectors' bandwidth can also be made by reducing the laser submount thickness to shorten the wirebonds.

Integrating these potential enhancements from both laser and detector should support higher symbol rates and modulation levels, which would potentially elevate the speed of room-temperature LWIR FSO links to near or even beyond 100 Gbit s$^{-1}$ on a single channel in the near term. In the longer term, this atmospheric transmission window targets long-distance applications. A preliminary link budget analysis is performed (see Supplementary Information Note 4). We foresee that extensive engineering efforts will be required to enhance the transmission distance to meet practical requirements, building upon the foundation laid by this work. Furthermore, facilitating non-line-of-sight (NLOS) transmission would benefit many terrestrial applications. Inspired by the reconfigurable metasurface concept proposed for THz

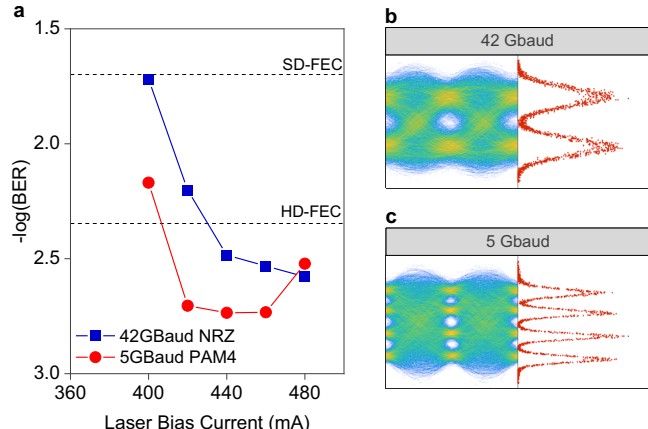

**Fig. 5 | Free-space transmission results using the RF-QCL and the QCD. a** BER versus the laser bias current for NRZ and PAM4 at the highest symbol rates achievable to meet the HD-FEC threshold. **b, c** Selected eye diagrams and the symbol distribution histograms for both modulation formats, measured at the highest bias current point after receiver equalization.

communication[58], intersubband polaritonic metasurfaces operating in the MIR could potentially enable NLOS FSO communication[59].

Finally, we acknowledge other technological alternatives for MIR FSO semiconductor transceivers. First, promising results have been demonstrated with interband cascade lasers (ICL)[49,60,61], primarily targeting the the MWIR window. Compared to QCL, ICL requires lower bias current, conceivably leading to reduced power consumption. Recently, up to 14 Gbit s$^{-1}$ PAM4 transmission has been demonstrated with a directly modulated Fabry–Perot ICL at 4.18 μm[61]. Lately, high-temperature (>200 K) ICLs operating in the LWIR window have been demonstrated[62], indicating their potential for LWIR FSO. Another approach is external modulation, which we believe offers more long-term potential for high-speed coherent LWIR FSO communications[33,63,64]. With an external Stark-effect modulator, over 20 Gbit s$^{-1}$ FSO transmission at 9 μm has been demonstrated using a room-temperature QCD and over 30 Gbit s$^{-1}$ using a nitrogen-cooled QWIP at 77 K[64]. The main challenges of this coherent technology roadmap include precise modulator light coupling and its associated power loss, independent phase modulation[65], and linear coherent reception. While MIR semiconductor transceiver technologies are still emerging, they've advanced significantly in the past decade. As they mature, they should be benchmarked using standard metrology for thorough performance comparisons.

## Methods
### Fabrication of the quantum cascade lasers
The design and operational principle of unipolar quantum optoelectonic devices, i.e., QCL, QCD and QWIP have been thoroughly studied (see details in Supplementary Information Note 1). For the two QCLs used in the experiment, both are distributed feedback lasers with length of 4 mm and they are fabricated by mirSense. Their active region consists of successive GaInAs wells and AlInAs barriers with a strained composition, grown using Molecular Beam Epitaxy (MBE) on an InP substrate. A buried geometry was chosen to meet the continuous wave (CW) operation requirement. In this configuration, InP:Fe regrowth was performed using hydride vapor phase epitaxy (HVPE) by KTH after ridge etching, reducing waveguide losses and thermal heating of the active region.

### Experimental configuration of the free-space transmission
The experimental setup is shown in Fig. 3a. The digital waveforms of signals with different modulation formats and symbol rates were generated offline with typical transmitter-side digital signal processing (DSP) routine that is widely used in fibre-optic datacom systems,

detailed in the next section. The generated digital samples are converted to the analog domain by an arbitrary waveform generator (AWG, Keysight M8195A) with a digital-to-analog converter (DAC) of 65 GSample s$^{-1}$ sampling rate, 8-bit resolution and a memory length of 16 GSamples. The output of the AWG is configured to be between 120 mV$_{peak-to-peak}$ (mV$_{pp}$) and 250 mV$_{pp}$, depending on the modulation formats and symbol rates. An electrical amplifier (SHF 804B) of 66 GHz bandwidth and 22 dB gain is used to amplify the signal to the range between 2 V$_{pp}$ and 3.1 V$_{pp}$. A high-current broadband bias-tee (Maki Microwave BT2-0040), which can handle up to 2 A DC current with an RF bandwidth of 40 GHz, is used for combining the modulation signal with a DC current before sending it to the QCL mount. The QCLs are mounted on Peltier element thermoelectric cooling (TEC) to stabilize the operational temperature to 15 °C for both lasers during all test cases. A pair of ZnSe Aspheric lenses (Thorlabs AL72512-E3) with a focal length of 12.7 mm are used to collimate the QCL output beam and focus the beam on the detectors at the receiver. In this experiment, due to the high precision requirement on focusing the beam with small spot size to the detectors (60 × 60 μm$^2$) to maximize the incident optical power, we limit the distance to 20 cm between the transmitter and the receiver. It is noted that this distance was chosen as an ease of implementation rather than an upper limit, as the beam collimation and focusing performance within the lab space (<10 m) is expected to be virtually the same since nearly all the emitting power can be collected and sent to the detector by careful collimation and focusing. When transmitting over longer distances, one challenge is the broadening of the beam waist for such long-wavelength signals. This broadening can lead to a reduction of received signal power due to the limitations of the receiver's aperture size. More analysis of beam broadending is described in Supplementary Information Note 4.

The receiver consists of a photodetector, a second electrical amplifier (SHF 804B) and a real-time digital storage oscilloscope (DSO, Keysight DSAZ334A) of 80 GSample s$^{-1}$. The photodetector used is either the QCD or the QWIP. The QCD operates entirely passively, requiring no bias voltage during operation. In contrast, the QWIP requires a second bias-tee to supply a 1.1 V bias voltage. After photodetection, the generated photocurrent is amplified and sampled before sending back to the lab computer for receiver DSP and performance evaluation.

### Bandwidth characterization on the device level and on the system level
Bandwidth characterisations of the unipolar optoelectronics were performed using the electrical rectification method[52] at the device-level. For the QCD and the QWIP, a DC voltage and a RF signal are combined at a bias-tee and delivered to the detectors over a 50 Ω impedance transmission line. The rectified DC current is generated when applying specific bias voltage so the devices' I-V characteristics are in the nonlinear region, which is propotional to the magnitude of the device transfer function at the AC frequency. The rectified DC current, which is in the order of a few μA, can be directly measured with the DC source (Keithley 2450 SourceMeter). Subsequently, sweeping the RF frequency enables the acquisition of the complete amplitude frequency response. For the QCL, which is an active device requiring a large DC current of hundreds of mA, an additional low-frequency signal (much lower than the RF frequency) is modulated on the RF signal. This produces a rectified current displaced from DC, detectable and analyzable with a lock-in amplifier. Similarly, by continuously measuring this rectified current while sweeping the RF frequency, we can obtain the overall amplitude modulation response of the QCL. Detailed insight into the operational mechanics of this electrical rectification method is available in Supplementary Information Note 2.

System-level bandwidth characteristisation is performed by generating flat frequency combs with the AWG, transmitting them through the whole system and capturing them with the DSO. In this way the end-to-end system frequency response can be obtained by comparing both

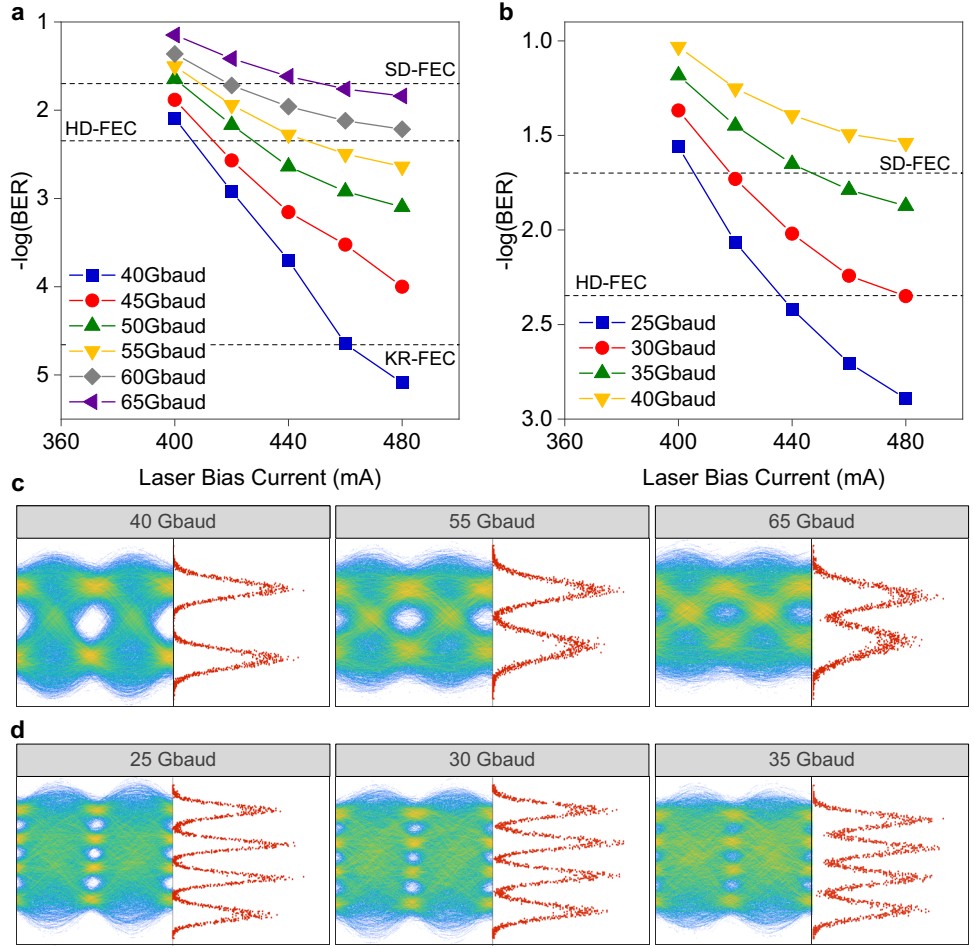

**Fig. 6 | Free-space transmission results using the RF-QCL and the QWIP. a** BER results for NRZ signals at different symbol rates, measured against different laser bias current points. The KR-FEC limit of $5.2 \times 10^{-5}$ BER is also shown as a benchmark. **b** BER results for PAM4 signals at different symbol rates, measured against different laser bias current points. **c**, **d** Selected eye diagrams and the symbol distribution histograms measured at highest bias current point after receiver equalization for NRZ and PAM4 at different symbol rates, respectively.

the amplitude and the phase of the frequency comb lines. The characterization results of all test cases are shown in Supplementary Fig. 7.

### Digital signal processing (DSP) at the transmitter and the receiver for data transmission

The DSP algorithms used in this experiment are standard routines developed for short-reach fibre-optic communications. The transmitter DSP consists of pseudo-random binary sequence (PRBS) generation, root-raise-cosine (RRC) pulse shaping, resampling to match the AWG sampling rate, and a static 2-tap pre-emphasis filter. The receiver DSP routine consists of a matched RRC filter, up-sampling and timing recovery, a data-aided decision feedback equalizer (DFE) with 99 feedforward taps and 99 feedback taps, symbol demodulation, and bit error rate (BER) counting. The number of DFE taps were fixed during all test cases for simplicity and maximizing the performance, which can be potentially reduced for some of the test cases. A comprehensive analysis of the effect of DFE taps on the BER performance is presented in Supplementary Figs. 8–11. These figures illustrate example test cases using RF-QCL in conjunction with QCD and QWIP, respectively. A block diagram and more details of the DSP configuration can be found in Supplementary Information Note 3.

### Calculation of net bitrates for different modulation formats

Calculating the net bitrates, expressed in Gbit s$^{-1}$, from the symbol rates, measured in Gbaud, consists of two steps. The first step involves converting the symbol rate to the gross bitrate. In the second step, we calculate the net bitrate from the gross bitrate by deducting the FEC overhead (OH).

In the first step, when utilizing single carrier signal formats such as those employed in this study, the formula for calculating the gross bitrate (Gbit s$^{-1}$) is: Gbit s$^{-1}$ = bits/symbol × Gbaud. Specifically, for a binary NRZ signal, the gross bitrate directly matches the baud rate; for instance, a 55 Gbaud NRZ signal corresponds to a gross bitrate of 55 Gbit s$^{-1}$. For M-level PAM signals, the calculation of bits/symbol is given by: bits/symbol = $\log_2(M)$. Consequently, in the case of PAM4, each symbol equates to 2 bits, resulting in a bitrate of Gbit s$^{-1}$ = 2 × Gbaud. For PAM6, where bits per symbol are approximately 2.585, we often round it to 2.5 for practical configurations to prevent the requirement for a long bit-to-symbol mapping memory.

In the subsequent step, the net bitrate is calculated by dividing the gross bitrate by (1 + FEC OH). The overhead for the hard-decision FEC threshold, which we evaluated in our study, is 6.25%. For the soft-decision FEC, which tolerates a higher BER, the overhead is 20%.

Summarizing both steps, the relation between symbol rate and net bitrate for M-level PAM signal is expressed as:

$$net\ bitrate = \frac{symbol\ rate \times \log_2(M)}{1 + FEC\ OH}$$

## Data availability
The measurement data generated in this study have been deposited in https://doi.org/10.5281/zenodo.12515593.

## Code availability
The algorithms used for the digital signal processing at the transmitter and the receiver are standard and are outlined in detail in the Methods and Supplementary Information. All codes of the DSP algorithms used in this study are embedded in a larger framework, which together with specific user instructions can be available from the corresponding authors upon request.

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

## Acknowledgements

This work was supported in part by the EU H2020 cFLOW Project (828893), in part by the Swedish Research Council (VR) project 2019-05197 and project 'BRAIN' 2022-04798, in part by the COST Action CA19111 NEWFOCUS, VINNOVA-funded project 'A-FRONTHAUL' 2023-00659, and in part by the LZP FLPP project 'MIR-FAST' (lzp-2023/1-0503). The authors from ENS acknowledge the financial support of the ENS-Thales Chair, Direction Générale de l'Armement (DGA), PEPR Electronique, ANR project LIGNEDEMIR (ANR- 18CE09-0035), FET Open projects cFLOW (Grant No. 828893) and CNRS Renatech network.

## Author contributions

H.D. and X.P. proposed the long-wave IR FSO transmission study with directly modulated QCL and QCD/QWIP detectors. H.D., M.J., L.D., R.S., and X.P. designed the overall experiment. I.S and K.P. provided the epitaxial growth for the detectors. L.D., G.M. and Y.-T. S. developed the QCL chips and submounts, and characterised the QCLs. H.D., T.B., D.G., A.V., and C.S. developed the QCD and QWIP chips, performed RF bounding and device characterisation. L.Z., X.Y., O.O., and X.P. developed the PAM modulation and DSP routine. T.S., S.S., and V.B. calibrated and performed test instrumentation, H.D., M.J., L.D., A.O., R.S., and X.P. carried out the FSO transmission experiment. M.J. and X.P. processed and analyzed the experimental data. R.S., G.M., D.G., Y.-T. S., O.O., and C.S. assisted in discussing and interpreting the results. X.P. and C.S. coordinated and supervised the experiment. H. D., M.J., L.D., and X.P. wrote the draft of the manuscript. All the authors reviewed and edited the manuscript.

## Funding

## Competing interests

The authors declare no competing interests.
