## [Peer Review File · Nature Communications]

Unipolar Quantum Optoelectronics for High Speed Direct Modulation and Transmission in 8-14 μm Atmospheric WindowEditorial note: This manuscript has been previously reviewed at another journal that is not operating a transparent peer review scheme. This document only contains reviewer comments and rebuttal letters for versions considered at *Nature Communications*.

REVIEWERS' COMMENTS

Reviewer #1 (Remarks to the Author):

In this new version, the authors emphasize that the system performance is excellent for FSO communications. However, I remain dissatisfied with the short distance between the transmitter and receiver, which is only 20 cm. The primary objective of FSO in the LWIR band is to enable long-distance communication. A distance of merely 20 cm is inadequate and not practical at this stage.

Reviewer #2 (Remarks to the Author):

Dear [name redacted]

Dely et al. present here a revised version of their manuscript on the exploitation of the so-called long-wave infrared (LWIR) spectral region ($\sim 8\text{-}14\ \mu\text{m}$) for free-space telecom applications demonstrating record-high bitrates beyond 55 Gbits-1. The manuscript has been transferred to Nature Communications and I will base my considerations for giving a recommendation on this fact together with their reply and revisions of their original manuscript.

From my point of view, the manuscript has improved significantly as compared to its initial version and all three important issues that I have raised in my first review have been addressed adequately, This includes: 1) giving a more detailed explanation of the novelty and impact of this work as compared to the authors previous results and other results from literature, 2) significantly revising and extending the literature section to give a better coverage of the state-of-the-art in the field and 3) adding much needed clarifications and explanations in certain parts of the manuscript.

Thus, I can conclude: Dely et al. show a significant advancement in the field of MIR (in particular the LWIR section of the MIR) free-space telecommunication systems with

significant novelties on the system and system performance level. They present a very strong technical work that is presented in a suitable way and which I believe is of important relevance to the photonics community in general and also well beyond. Thus it qualifies to be of significant interest and impact to the (multidisciplinary) audience of Nature Communications. My assessment therefore leads to my recommendation to accept the manuscript for publication in Nature Communications in its present form.

Best regards,

Reviewer 2

Response letter for manuscript NCOMMS-24-30909A entitled
"Unipolar Quantum Optoelectronics for High Speed Direct Modulation and Transmission in 8-
14 μm Atmospheric Window"

Reviewers' Comments

Reviewer: 1

In this new version, the authors emphasize that the system performance is excellent for FSO communications. However, I remain dissatisfied with the short distance between the transmitter and receiver, which is only 20 cm. The primary objective of FSO in the LWIR band is to enable long-distance communication. A distance of merely 20 cm is inadequate and not practical at this stage.

Authors' reply: We appreciate and understand your opinion. We fully acknowledge the importance of transmission distance for FSO communications. Extensive future engineering efforts will be required to enhance the transmission distance to meet practical requirements, building upon the foundation laid by this work.

Changes in the manuscript:

In Discussion, page 17 line 256-258

We foresee that extensive engineering efforts will be required to enhance the transmission distance to meet practical requirements, building upon the foundation laid by this work.

Reviewer: 2

Dely et al. present here a revised version of their manuscript on the exploitation of the so-called long-wave infrared (LWIR) spectral region ($\sim 8\text{-}14\ \mu\text{m}$) for free-space telecom applications demonstrating record-high bitrates beyond 55 Gbits-1. The manuscript has been transferred to Nature Communications and I will base my considerations for giving a recommendation on this fact together with their reply and revisions of their original manuscript.

From my point of view, the manuscript has improved significantly as compared to its initial version and all three important issues that I have raised in my first review have been addressed adequately, This includes: 1) giving a more detailed explanation of the novelty and impact of this work as compared to the authors previous results and other results from literature, 2) significantly revising and extending the literature section to give a better coverage of the state-of-the-art in the field and 3) adding much needed clarifications and explanations in certain parts of the manuscript.

Thus, I can conclude: Dely et al. show a significant advancement in the field of MIR (in particular the LWIR section of the MIR) free-space telecommunication systems with significant novelties on the system and system performance level. They present a very strong technical work that is presented in a suitable way and which I believe is of important relevance to the photonics community in general and also well beyond. Thus, it qualifies to be of significant interest and impact to the (multidisciplinary) audience of Nature Communications. My assessment therefore leads to my recommendation to accept the manuscript for publication in Nature Communications in its present form.

Authors' reply: We really appreciate your positive feedback recognizing the novelty and importance of our work.